# When Is Rank-1 Enough? Geometry-Guided Initialization for Parameter-Efficient Fine-Tuning

Haoran Zhao [1]   Soyeon Caren Han [1]   Eduard Hovy [1]

## Abstract

Parameter-efficient fine-tuning (PEFT) is a standard way to adapt multimodal large language models, yet extremely low-rank settings—especially rank-1 LoRA—are often unstable. We show that this instability is not solely due to limited capacity: in the rank-1 regime, optimization is highly sensitive to the *update direction*. Concretely, pretrained vision and text features form mismatched anisotropic regions, yielding a dominant "gap" direction that acts like a translation component and disproportionately steers early gradients under rank-1 constraints. Analyzing pretrained representations, we identify a modality-gap axis that dominates early gradient flow, while a random rank-1 initialization is unlikely to align with it, leading to weak gradients and training collapse. We propose **Gap-Init**, a geometry-aware initialization that aligns the rank-1 LoRA direction with an estimated modality-gap vector from a small calibration set, while keeping the initial LoRA update zero. Across multiple vision-language tasks and backbones, Gap-Init consistently stabilizes rank-1 training and can match or outperform strong rank-8 baselines. Our results suggest that at the extreme low-rank limit, *initial alignment can matter as much as rank itself*. Code is available at https://github.com/HaoranZhao2000/Gap-Init.

## 1. Introduction

Large multimodal language models (MLLMs) have achieved strong performance on vision–language tasks such as image captioning and visual question answering (Alayrac et al., 2022; Li et al., 2022; 2023). However, adapting these models to downstream tasks remains computationally expensive, motivating parameter-efficient fine-tuning (PEFT) methods such as LoRA (Houlsby et al., 2019; Hu et al., 2022b). By restricting updates to low-rank adapters, PEFT reduces training cost while often preserving performance. Despite their success, low-rank settings, particularly rank-1 adaptation, are often unstable in practice, commonly attributed to limited expressive capacity or poorly matched initialization (Hu et al., 2022b; Dettmers et al., 2023; Meng et al., 2024). As a result, increasing rank is typically treated as the primary remedy for improving stability and accuracy.

In this work, we show that capacity alone does not fully account for the instabilities observed at the rank-1 limit in modern multimodal learning settings. Across several widely used MLLM architectures, we find that rank-1 optimization can be disproportionately influenced by whether the update direction has non-trivial projection onto a geometry-salient cross-modal axis in activation space. Under rank-1 constraints, a randomly initialized direction is likely to have negligible projection onto such an axis, which can weaken early gradient signal and lead to optimization failure. This observation aligns with known properties of high-dimensional representations, where random directions concentrate near orthogonality to fixed semantic axes in practice (Ethayarajh, 2019; Mu & Viswanath, 2018a; Vershynin, 2018).

Importantly, we do not claim that a single rank-1 direction captures all task-relevant variation. Rather, our evidence suggests that in the extreme low-rank regime, in many practical settings, early optimization can be dominated by alignment with a small number of geometry-salient axes, even if these axes explain only a modest fraction of representational variance. Motivated by this insight, we propose **Gap-Init**, a simple, training-free initialization strategy that aligns rank-1 updates with an empirically estimated cross-modal gap direction obtained from a small calibration set, while keeping the LoRA update zero at initialization.

Across multiple multimodal benchmarks, Gap-Init improves the stability and effectiveness of rank-1 LoRA, and in several settings it approaches the performance of higher-rank baselines with substantially fewer trainable parameters. Overall, our results highlight that at the extreme low-rank limit, initialization and representation geometry can be first-order determinants of trainability, complementing capacity-

[1]School of Computing and Information Systems, University of Melbourne, Melbourne, Australia. Correspondence to: Soyeon Caren Han <caren.han@unimelb.edu.au>.

*Proceedings of the $43^{rd}$ International Conference on Machine Learning*, Seoul, South Korea. PMLR 306, 2026. Copyright 2026 by the author(s).

based perspectives on PEFT from a geometric standpoint.

Our contributions are summarized as follows:

- **A geometric view of rank-1 instability:** we identify a systematic and intrinsic optimization barrier in rank-1 PEFT where random low-rank updates can be nearly orthogonal to a salient cross-modal axis, suppressing early gradient flow and destabilizing training.

- **Gap-Init:** we introduce a lightweight, calibration-based initialization that aligns the rank-1 LoRA direction with an estimated modality-gap vector while preserving the pretrained function at initialization.

- **Empirical evidence across tasks/backbones:** we show that Gap-Init enables stable and effective rank-1 adaptation on multiple multimodal benchmarks, and is often competitive with higher-rank baselines under the same standard training protocol.

## 2. Related Work

**Parameter-efficient fine-tuning (PEFT).** LoRA learns low-rank updates while freezing pretrained weights, substantially reducing trainable parameters with competitive performance (Hu et al., 2022b). QLoRA combines low-rank adapters with 4-bit quantization for memory-efficient fine-tuning (Dettmers et al., 2023). Beyond standard LoRA, recent methods improve low-rank adaptation by exploiting structure in pretrained weights: DoRA decomposes weights into magnitude and directional components and applies low-rank updates in the decomposed form (Liu et al., 2024), while PiSSA replaces the standard initialization with an SVD-based spectral initialization derived from the pretrained weight matrix (Meng et al., 2024). These approaches primarily operate in *weight space* and can improve optimization and generalization under moderate ranks. A common assumption underlying these methods is that task-relevant adaptation directions can be inferred directly from the pretrained weight geometry, without explicit reference to data-induced activation patterns.

**PEFT for vision–language models.** Adapter- and LoRA-style tuning are widely used for multimodal transfer. VL-Adapter injects lightweight adapters to approach full fine-tuning performance on vision–language tasks while updating only a small fraction of parameters (Sung et al., 2022). Large-scale MLLMs such as BLIP-2 and LLaVA commonly rely on adapter/LoRA modules during adaptation (Li et al., 2023; Liu et al., 2023). However, most prior multimodal PEFT studies focus on moderate ranks (e.g., $r \geq 8$) and do not specifically analyze the instability of standard LoRA initialization in the extreme rank-1 regime, where optimization becomes sensitive to initialization geometry in practice.

*Table 1.* **Rank-1 adaptation: weight-space vs. activation-space.** Representative PEFT refinements (PiSSA (Meng et al., 2024), DoRA (Liu et al., 2024)) focus on *weight-space* parameterization, while Gap-Init considers *activation-space* geometry estimated from data for rank-1 adaptation in practice.

| Method | Geometry | Data | Rank-1 Focus |
|--------|----------|------|--------------|
| PiSSA | Weight | No | No |
| DoRA | Weight | No | No |
| Gap-Init | Activation | Yes | Yes |

**Representation geometry and modality gap.** Neural embeddings are known to be highly anisotropic: early analyses show that removing dominant components can improve isotropy and downstream performance (Mu & Viswanath, 2018b), and contextual representations exhibit strong anisotropy and layer-dependent geometry (Ethayarajh, 2019). In multimodal encoders, Liang et al. (2022) formalize the *modality gap*, where image and text embeddings occupy separated regions in a shared space.

**Positioning of our approach.** Gap-Init is complementary to weight-space PEFT refinements (e.g., DoRA, PiSSA): rather than selecting subspaces by diagonalizing the pretrained weights, we leverage *data-induced activation-space* geometry by estimating a cross-modal gap direction from a small calibration set and using it to initialize rank-1 adaptation. This perspective is particularly relevant when optimization is constrained to a single update direction.

## 3. The Geometry of Modality Alignment

We characterize a simple geometric structure in pretrained MLLMs that is practically relevant for extreme low-rank adaptation. Empirically, (i) vision and text hidden states are highly anisotropic and concentrate in narrow cones, and (ii) their cross-modal mismatch admits a prominent mean-shift (translation-like) component. Under rank 1 constraints, optimization becomes highly sensitive to whether the update direction has non-trivial projection onto this gap axis; a random rank-1 direction is likely to be nearly orthogonal in high dimensions, attenuating the effective gradient signal. Gap-Init leverages this geometry by initializing the rank-1 LoRA direction to align with an estimated gap vector.

### 3.1. Why Random Rank-1 Initialization Struggles: Concentration Near Orthogonality

Let $u = g/\|g\|$ denote the normalized modality-gap direction. Standard LoRA initialization induces (up to scaling) a random rank-1 direction $b$ that is approximately uniform on $\mathbb{S}^{d-1}$. In high dimensions, random vectors are overwhelmingly likely to be nearly orthogonal to any fixed direction, *with high probability*. The following proposition formalizes this concentration phenomenon.

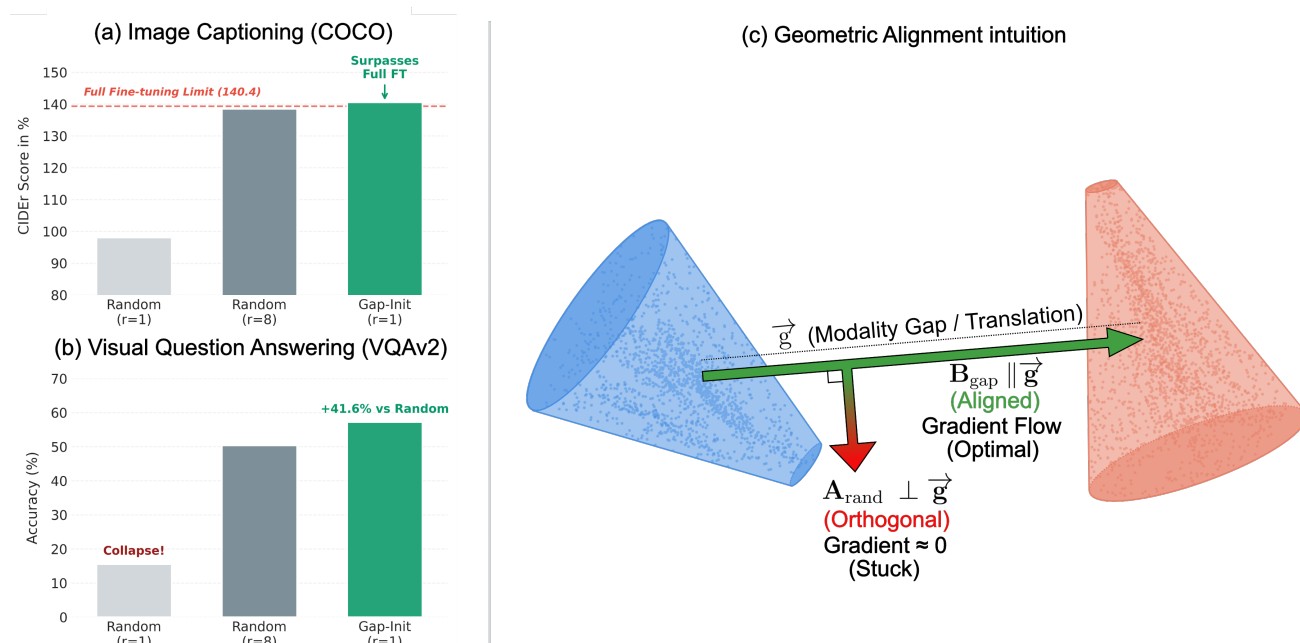

*Figure 1.* **Gap-Init stabilizes rank-1 adaptation via geometry-aware initialization across multimodal tasks and training settings. (a) Image Captioning (COCO).** Rank-1 Gap-Init matches or slightly exceeds strong rank-8 baselines in this setting while using substantially fewer trainable parameters. **(b) Visual Question Answering (VQAv2).** Gap-Init improves rank-1 accuracy over random initialization and yields stable convergence. **(c) Geometric intuition.** Vision and text embeddings form anisotropic cones; their mismatch admits a translation-like component that is particularly consequential for optimization at rank 1. Random rank-1 initialization typically has negligible projection onto this axis, suppressing useful gradient flow. Gap-Init aligns the update direction with the estimated gap vector, improving optimization dynamics from the first step and throughout training across tasks.

**Proposition 3.1** (Concentration of random directions). *If $b$ is uniformly random on $\mathbb{S}^{d-1}$ and $u$ is fixed, then*

$$\mathbb{E}[\langle u, b \rangle] = 0, \qquad \mathbb{E}[\langle u, b \rangle^2] = \frac{1}{d},$$

*and for any $\varepsilon > 0$,*

$$\mathbb{P}(|\langle u, b \rangle| > \varepsilon) \leq 2 \exp\left(-\frac{(d-1)\varepsilon^2}{2}\right).$$

For BLIP-2 ($d = 4096$), this implies $|\langle u, b \rangle|$ is typically on the order of $1/\sqrt{d}$ in high dimensions.

Fig. 2 shows the empirical distribution of the initial alignment: random rank-1 directions cluster sharply around cosine $\approx 0$, while Gap-Init achieves high alignment (e.g., $\cos \approx 0.97$ in our setting). Consequently, the component of the update (and thus early gradients) along the gap axis can be strongly attenuated under random rank-1 initialization.

### 3.2. An Optimization-Relevant Translation Component in the Modality Gap

If cross-modal alignment required coordinating many independent directions, rank-1 adaptation would be unlikely to provide a reliable optimization signal in practice. Instead, the singular-value spectrum of per-sample gap vec-

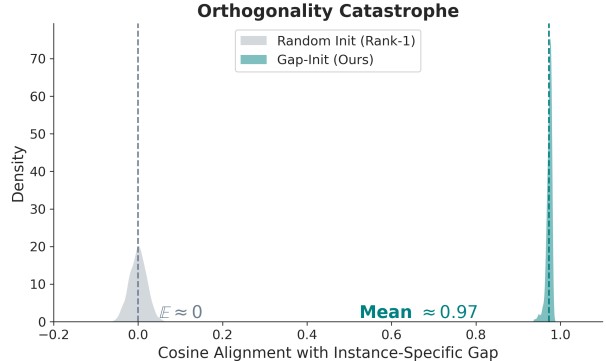

*Figure 2.* **Near-orthogonality at initialization.** Random rank-1 LoRA directions concentrate near $\cos(u, b) \approx 0$, whereas Gap-Init aligns the direction with the estimated gap axis by construction.

tors (Fig. 3) indicates a structured mismatch: (i) the leading direction captures a non-negligible fraction of the gap variation (e.g., $\sim$16% in our analysis), and (ii) subsequent components contribute progressively less.

This motivates modeling the mismatch with a translation-like component shared across samples:

$$T(f_{\text{vision}}(x)) \approx f_{\text{vision}}(x) + \lambda(x)\,g,$$

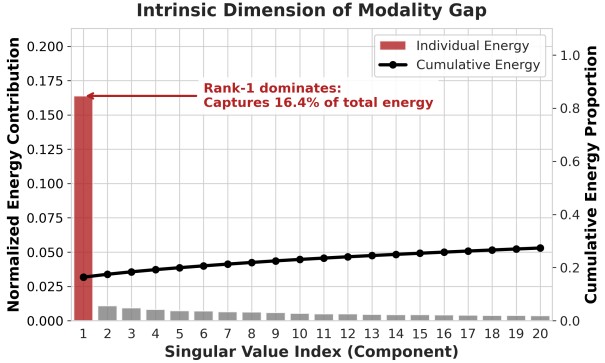

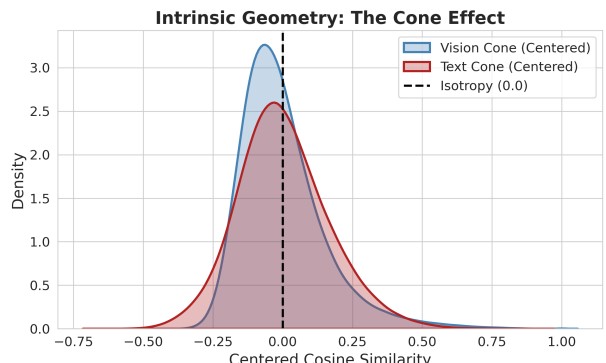

*Figure 3.* **Spectrum of instance-specific gap vectors.** The singular-value spectrum suggests a prominent leading direction, with diminishing contributions from higher-rank components.

*Figure 4.* **Intrinsic geometry (cone effect).** After centering, both vision and text embeddings remain strongly concentrated around their mean directions, indicating anisotropy within each modality.

where $g$ is a global gap direction and $\lambda(x)$ is a sample-dependent scalar. While multimodal alignment is not intrinsically one-dimensional, under rank 1 constraints the availability of gradient signal can be dominated by alignment with such a prominent direction during early optimization. Gap-Init operationalizes this observation by aligning the rank-1 LoRA subspace with $g$ at initialization.

A formal analysis via a Gaussian translation model with synthetic validation is provided in Appendix A, while additional discussion on when translation-like structure can emerge appears in Appendix B.

### 3.3. The Cone Hypothesis: Modality Clusters Are Mean-Dominated

Prior work on embedding anisotropy (Mu & Viswanath, 2018b; Ethayarajh, 2019) and multimodal contrastive models (Liang et al., 2022) shows that high-dimensional representations often concentrate in narrow angular regions (*cones*). We confirm this behavior in BLIP-2 by measuring the cosine similarity between centered vision/text embeddings and their respective mean vectors.

As shown in Fig. 4, both modalities exhibit tight concentration, consistent with mean-dominated geometry. This motivates modeling the cross-modal mismatch as a dominant translation-like direction, for which the difference between modality means provides a simple estimator:

$$g = \mu_{\text{text}} - \mu_{\text{vision}}.$$

In what follows, we study how alignment with this gap direction affects optimization when the adaptation subspace is restricted to rank 1 in this setting.

## 4. Methodology

Motivated by the geometric analysis in Section 3, we propose **Gap-Init**, a geometry-aware initialization strategy that rescues rank-1 adaptation by explicitly aligning the LoRA update direction with the modality gap. Crucially, our approach is *layer-specific*: the modality gap is estimated *within the hidden-state space of each transformer layer*, and LoRA parameters at that layer are initialized using the corresponding gap vector. This design ensures strict dimensional and coordinate consistency and avoids relying on a single global direction across heterogeneous layers.

Our method consists of three components: (1) revisiting the failure mode of rank-1 low-rank adaptation, (2) estimating a translation-dominant modality gap in aligned representation space, and (3) initializing LoRA parameters to align with this gap. We further validate the diagnostic value of the gap through a gap-guided layer selection strategy.

### 4.1. Preliminaries: Rank-1 Adaptation and Its Limitations

Let $W_0 \in \mathbb{R}^{d \times k}$ denote a frozen pretrained weight matrix of a transformer layer. LoRA (Hu et al., 2022a) parameterizes the trainable update as

$$\Delta W = BA,$$

where $B \in \mathbb{R}^{d \times r}$ and $A \in \mathbb{R}^{r \times k}$ with $r \ll \min(d, k)$. The forward computation becomes

$$h = (W_0 + \Delta W)x = W_0 x + BAx. \qquad (1)$$

Following standard practice, $B$ is initialized to zero and $A$ is initialized from a Gaussian distribution.

For moderate ranks ($r \geq 8$), the optimizer can explore multiple directions and gradually align the update subspace with task-relevant geometry. At rank 1, however, the trainable

subspace collapses to a single line determined entirely by the initialization. As shown in Section 3, in high-dimensional spaces ($d \approx 10^3$), a randomly initialized direction is almost surely orthogonal to the modality-gap direction. Consequently, the gradient component along the useful direction is suppressed by a factor of $O(d^{-1/2})$, leading to unstable optimization or complete collapse—a phenomenon we refer to as the *orthogonality catastrophe*.

### 4.2. Estimating the Translation-Dominant Modality Gap

Section 3 shows that the dominant discrepancy between vision and text representations manifests as a translation between their respective representation cones. We estimate this translation using a small, unlabeled calibration set $\mathcal{D}_{\mathrm{cal}}$ and a frozen pretrained model. The calibration data do not require exact supervision or precise alignment; weakly paired image–text examples are sufficient to recover the dominant translation direction in the representation space.

**Aligned representation space.** Visual features are first mapped into the language model's semantic space using the vision–language alignment module (e.g., Q-Former followed by a linear projection). As a result, both visual and textual representations reside in the same hidden-state space of the language model.

**Layer-wise gap estimation.** For each transformer layer $\ell$, and for each calibration sample $(x_{\mathrm{img}}^{(i)}, x_{\mathrm{txt}}^{(i)})$, we extract:

$$z_v^{(i,\ell)} = h_v^{(\ell)}(x_{\mathrm{img}}^{(i)}), \qquad z_t^{(i,\ell)} = h_t^{(\ell)}(x_{\mathrm{txt}}^{(i)}),$$

where $h_v^{(\ell)}$ and $h_t^{(\ell)}$ denote the aligned visual and textual hidden states at layer $\ell$.

We define the *sample-projected gap* at layer $\ell$ as

$$g^{(i,\ell)} = z_t^{(i,\ell)} - z_v^{(i,\ell)}.$$

**SPOT-Gap.** To reduce noise and improve robustness, we optionally stabilize these vectors using local neighborhood consistency (SPOT-Gap; details in Appendix C). The final *layer-specific modality gap* is computed as

$$g^{(\ell)} = \frac{1}{|\mathcal{D}_{\mathrm{cal}}|} \sum_i \tilde{g}^{(i,\ell)}.$$

This procedure requires a single forward pass over the calibration set and incurs negligible computational overhead.

### 4.3. Gap-Init: Geometry-Aware Initialization

Gap-Init initializes LoRA parameters such that the update subspace at each layer is aligned with the corresponding modality gap. Importantly, the gap vector and the LoRA parameters are defined in the *same layer-specific hidden space*, ensuring dimensional and coordinate consistency.

For rank 1, the LoRA matrices reduce to vectors accordingly. At layer $\ell$, we initialize

$$B^{(\ell)} \leftarrow \frac{g^{(\ell)}}{\|g^{(\ell)}\|_2}, \qquad A^{(\ell)} \leftarrow 0.$$

Setting $A^{(\ell)} = 0$ ensures $\Delta W^{(\ell)} = 0$ at initialization, thereby preserving the pretrained model's behavior.

Although the forward pass is unchanged, the backward dynamics are fundamentally altered. The gradient with respect to $A^{(\ell)}$ becomes

$$\frac{\partial \mathcal{L}}{\partial A^{(\ell)}} = \left(B^{(\ell)}\right)^\top \frac{\partial \mathcal{L}}{\partial h^{(\ell)}} x^\top,$$

which directly projects gradients onto the modality-gap direction itself. In contrast, standard LoRA must first discover this direction through optimization, which is ineffective under rank-1 constraints.

For higher ranks ($r > 1$), we align the first column of $B^{(\ell)}$ with $g^{(\ell)}$ and initialize the remaining columns randomly.

Notably, Gap-Init does not introduce new parameters, objectives, or training stages—it only changes the initial direction of an existing rank-1 adapter.

### 4.4. Gap-Guided Layer Selection (GG-Safe)

Although Gap-Init performs strongly even with naive layer selection, we further examine the diagnostic value of the modality gap by using it to select adaptation layers.

We compute a *layer-wise Spot-Gap score* $S_{\mathrm{gap}}^{(l)}$ that quantifies the discrepancy between visual and textual hidden representations at layer $l$ in this case. The score measures the magnitude of the average sample-projected gap in that layer (details in Appendix C).

We apply two constraints:

1. **Safety:** We exclude the bottom layers of the vision encoder, which encode low-level modality-specific features and should remain unchanged.

2. **Diagnosis:** Among the remaining layers, we select the top-$k$ layers with the highest $S_{\mathrm{gap}}^{(l)}$ values.

Empirically in Section 5, GG-Safe significantly outperforms naive selection under random initialization, validating the diagnostic relevance of the spot-gap metric. However, the proposed Gap-Init is sufficiently robust that its gains persist even without GG-Safe in practice.

**Algorithm 1** Gap-Init: Geometry-Aware Initialization for Rank-1 Adaptation

1: **Input:** Pretrained model with vision encoder $f_V$, text encoder $f_T$, projection module $\phi$; calibration set $\mathcal{D}_{\text{cal}}$; LoRA rank $r$.
2: **Output:** Initialized LoRA matrices $(B, A)$.
3: **Step 1: Extract aligned embeddings**
4: **for** each sample $(x_{\text{img}}, x_{\text{txt}}) \in \mathcal{D}_{\text{cal}}$ **do**
5:    $z_v \leftarrow \phi(f_V(x_{\text{img}}))$
6:    $z_t \leftarrow f_T(x_{\text{txt}})$
7:    $g^{(i)} \leftarrow z_t - z_v$    (sample-projected gap)
8: **end for**
9: **Step 2: Estimate layer-wise modality gaps (SPOT-Gap)**
10: **for** each layer $\ell$ **do**
11:    Apply SPOT-Gap smoothing to $\{g^{(i,\ell)}\}$
12:    $g^{(\ell)} \leftarrow \dfrac{1}{|\mathcal{D}_{\text{cal}}|} \sum_i \tilde{g}^{(i,\ell)}$
13: **end for**
14: **Step 3: Initialize LoRA matrices**
15: Initialize $B$ and $A$ with the desired shapes
16: Set the first column of $B^{(\ell)}$: $B^{(\ell)}_{:,1} \leftarrow g^{(\ell)}/\|g^{(\ell)}\|_2$
17: **if** $r > 1$ **then**
18:    Initialize the remaining columns of $B$ randomly
19: **end if**
20: $A \leftarrow 0$    (keeps $\Delta W = 0$ at initialization)
21: **return** $(B, A)$

## 5. Experiments

We evaluate Gap-Init across captioning (COCO), visual question answering (VQAv2, OK-VQA), zero-shot transfer (Flickr30k), and hallucination robustness (POPE), with all scores reported in percentage form.

### 5.1. Experimental Setup

**Model.** All experiments use BLIP-2 OPT-2.7B (Li et al., 2023), as it provides a clean and widely adopted architecture that decouples the vision encoder and the language model, making it particularly suitable for controlled analysis of cross-modal representations.

**Training and evaluation.** Models are fine-tuned on COCO Captioning (Karpathy split) and VQAv2, and evaluated on Flickr30k and OK-VQA for zero-shot generalization, as well as POPE for hallucination robustness.

**Baselines.** We compare against: (1) *Standard LoRA* with random initialization, (2) *PiSSA* (Meng et al., 2024), an SVD-based weight initialization method, and (3) *Rank-8 LoRA* and *Full-layer LoRA* as capacity upper bounds.

**Protocol.** Our primary focus is the rank-1 regime, represent-

*Table 2.* **COCO Captioning (Karpathy split).** Metrics are reported in percentage. Gap-Init at rank 1 achieves performance comparable to, or exceeding, stronger rank 8 baselines under the same training protocol.

| Method | Rk/Config | CIDEr | BLEU-4 |
|---|---|---|---|
| Standard LoRA | 1 / Naive | 98.08 | 24.48 |
| PiSSA | 1 / Naive | 130.42 | 39.07 |
| Standard LoRA | 8 / Naive | 138.49 | 40.63 |
| Full-layer LoRA | 1 / All | 140.36 | 41.65 |
| **Gap-Init (Ours)** | **1 / Naive** | **140.59** | **41.87** |

*Table 3.* **Zero-shot transfer to Flickr30k.** Metrics are reported in percentage. Gap-Init at rank 1 achieves the strongest zero-shot transfer performance among all rank-1 methods under the same training protocol and evaluation setting.

| Method | Rank | CIDEr | $\Delta$ vs Baseline |
|---|---|---|---|
| Standard LoRA | 1 | 63.00 | – |
| Standard LoRA | 8 | 78.35 | +15.35 |
| PiSSA | 1 | 76.00 | +13.00 |
| **Gap-Init (Ours)** | **1** | **79.60** | **+16.60** |

ing the practical limit of parameter efficiency in this study. Unless stated otherwise, we use naive layer selection (top-6 layers) and freeze the Q-Former.

### 5.2. COCO Captioning Results

On COCO captioning, rank-1 standard LoRA exhibits severe optimization instability, yielding a CIDEr score of 98.08 (Table 2). By contrast, Gap-Init stabilizes training and achieves 140.59 CIDEr, remaining competitive with rank-8 LoRA in this setting while using $8\times$ fewer trainable parameters, and comparable to full-layer LoRA on COCO captioning. This suggests that proper alignment of the update direction can be sufficient to recover the minimal subspace required for multimodal alignment.

### 5.3. Zero-Shot Transfer to Flickr30k

In zero-shot transfer to Flickr30k, Gap-Init achieves 79.60 CIDEr, outperforming other rank-1 methods and remaining competitive with rank-8 LoRA under the same training protocol (Table 3) under identical experimental conditions in evaluation. While PiSSA performs competitively in-domain, its transfer performance is lower (76.00 CIDEr), suggesting that weight-centric initialization may be more closely tied to pre-trained weight geometry, whereas data-dependent alignment can better support domain-agnostic transfer.

*Table 4.* **VQA performance and hallucination robustness (POPE).** Metrics are reported in percentage. Gap-Init stabilizes rank-1 training and achieves strong VQA performance while maintaining high POPE faithfulness.

| Method | Rank | VQA2 | OKVQA | POPE F1 |
|---|---|---|---|---|
| Standard LoRA | 1 | 15.58 | 2.93 | 50.1 |
| Standard LoRA | 8 | 50.39 | 17.02 | **94.8** |
| PiSSA | 1 | **63.35** | **26.56** | 86.9 |
| **Gap-Init (Ours)** | **1** | 57.23 | 21.03 | 92.4 |

## 5.4. Robustness and Reasoning: VQA and POPE

Gap-Init substantially improves rank-1 VQAv2 performance ($15.58 \rightarrow 57.23$) and yields stable convergence (Table 4). In terms of VQA accuracy, PiSSA attains higher scores under our training protocol, suggesting that weight-space spectral initialization can be effective for in-domain task performance. By contrast, Gap-Init achieves a higher POPE F1 (92.4 vs. 86.9), reflecting improved faithfulness under hallucination-focused evaluation. These results suggest that activation-space gap alignment provides a complementary benefit at rank 1, especially for stability and robustness-related metrics, relative to weight-space initialization methods. We emphasize that these comparisons focus on the extreme low-rank regime (rank 1), where optimization stability becomes a dominant factor.

## 6. Ablation and Analysis

### 6.1. Initialization vs. Layer Selection

We compare the effects of layer selection and initialization strategy under the rank-1 adaptation setting. As shown in Table 5, using naive layer selection with random initialization yields a baseline performance of 98.08 CIDEr. When GG-Safe is adopted for layer selection while keeping the initialization random, the performance improves modestly to 102.15 CIDEr in this setting.

When combining GG-Safe layer selection with Gap-Init, the performance increases substantially to 140.06 CIDEr. Further applying Gap-Init under naive layer selection achieves a slightly higher performance of **140.59 CIDEr**. The small gap between the two Gap-Init settings suggests that once the initialization direction is properly aligned, the specific choice of adaptation layers becomes less critical.

These results indicate that while GG-Safe layer selection contributes to rank-1 adaptation, the dominant performance gain originates from correcting the initialization direction. Under extremely low-rank constraints, optimization is primarily limited by the geometric alignment of the initialization rather than the particular choice of adaptation layers.

Further analyses on the robustness of the initialization di-

*Table 5.* **Initialization vs. Layer Selection under Rank-1 Adaptation.** Metrics are reported in percentage. Results show that while informed layer selection provides modest benefits, the dominant performance gain arises from correcting the initialization direction.

| Method | CIDEr |
|---|---|
| Naive Selection + Random Init | 98.08 |
| GG-Safe Selection + Random Init | 102.15 |
| GG-Safe Selection + Gap-Init | 140.06 |
| Naive Selection + Gap-Init | **140.59** |

*Table 6.* **Comparison with DoRA under low-rank adaptation.** Metrics are reported in percentage. Gap-Init at rank 1 remains competitive with DoRA across evaluated captioning metrics under the same training protocol, despite using a lower adaptation rank.

| Method | CIDEr | BLEU-4 |
|---|---|---|
| DoRA (r=1) | 140.51 | 41.55 |
| DoRA (r=8) | 139.55 | 40.92 |
| Gap-Init (r=1) | **140.59** | **41.81** |

rection, including the effects of calibration set size, data composition, and directional noise perturbations, are provided in Appendix D and Appendix E.

### 6.2. Comparison Against Stronger PEFT Baselines

We further compare Gap-Init (rank 1) with DoRA baselines following their default recommended configurations.

Table 6 compares the proposed Gap-Init with DoRA baselines under different ranks. DoRA with rank 1 achieves 140.51 CIDEr and 41.55 BLEU-4, while increasing the rank to 8 does not lead to additional gains.

Notably, Gap-Init with only rank 1 achieves the best overall performance, reaching 140.59 CIDEr and 41.81 BLEU-4 and surpassing both DoRA settings. These results indicate that increasing rank alone does not guarantee improved performance and that appropriate initialization can be more critical than adaptation capacity, making Gap-Init a parameter-efficient alternative to existing PEFT baselines.

### 6.3. Multi-Seed Evaluation

To rigorously evaluate the optimization stability of rank-1 adaptation, we compare Gap-Init with Standard LoRA across five distinct random seeds. The statistical summary is reported in Table 7.

The results in Table 7 reveal a critical vulnerability in standard low-rank adaptation. Under random initialization, Standard LoRA ($r = 1$) suffers from severe optimization instability, evidenced by a high standard deviation of 7.10 in CIDEr scores. While some seeds achieve competitive performance, others suffer from catastrophic collapse, effectively

*Table 7.* **Multi-seed Stability Analysis ($r = 1$).** We report Mean $\pm$ Standard Deviation across 5 seeds. Standard LoRA exhibits high variance and instability, whereas Gap-Init guarantees robust convergence with significantly reduced variance.

| Method | CIDEr | BLEU-4 |
|---|---|---|
| Standard LoRA ($r = 1$) | $135.37 \pm 7.10$ | $40.54 \pm 1.28$ |
| Gap-Init ($r = 1$) | $\mathbf{140.57 \pm 1.44}$ | $\mathbf{41.52 \pm 0.73}$ |

*Table 8.* **Cross-Backbone Generalization under Rank-1 Adaptation.** Metrics are reported in percentage. Gap-Init consistently improves performance over random initialization across vision–language backbones with different architectures and scales.

| Method | CIDEr | BLEU-4 | METEOR |
|---|---|---|---|
| **Qwen2-VL-7B-Instruct** | | | |
| Rank=1, Random | 143.67 | 42.55 | 31.89 |
| Rank=1, Gap-Init | **144.11** | **42.97** | **31.95** |
| **Gemma3-4B** | | | |
| Rank=1, Random | 90.35 | 24.95 | 24.28 |
| Rank=1, Gap-Init | **95.24** | **27.27** | **26.33** |

rendering the training process a lottery.

In sharp contrast, Gap-Init demonstrates superior robustness. By aligning the update direction with the modality gap, our method achieves a consistently high mean performance of 140.57 while reducing the standard deviation by nearly 5 times. This drastic reduction in variance confirms that Gap-Init eliminates the "orthogonality risk" inherent in high-dimensional random initialization, ensuring that rank-1 adaptation reliably converges to a high-performance solution regardless of the random seed.

### 6.4. Cross-Model Generalization Across Vision–Language Backbones

We evaluate the generalization of Gap-Init across vision–language backbones with different architectures and scales, including Qwen2-VL-7B-Instruct (Wang et al., 2024) and Gemma3-4B (DeepMind, 2025), which differ substantially in design. A summary of the results is provided in Table 8.

On the large-scale backbone Qwen2-VL-7B, rank-1 adaptation already performs competitively, with Gap-Init yielding consistent improvements across metrics. On the smaller-scale backbone Gemma3-4B, the advantage of Gap-Init becomes more pronounced under constrained optimization. With limited training budget, Gap-Init notably outperforms both rank-1 and rank-8 random initialization, demonstrating that aligning low-rank updates with dominant cross-modal discrepancies improves optimization efficiency. Further additional cross-backbone results under extended training budgets are provided in Appendix F.

### 6.5. Summary of Findings

Across ablation studies, Gap-Init demonstrates advantages in effectiveness, robustness, and parameter efficiency across diverse experimental conditions. First, it achieves optimal performance under a compact calibration regime, highlighting strong data efficiency. Second, investigations into calibration composition and noise injection reveal a nuanced geometric trade-off: while the modality gap is strictly domain-specific (as evidenced by the failure of OOD calibration), introducing compatible diversity—such as mixed-domain data or controlled noise—acts as beneficial regularization. Third, multi-seed evaluation shows that Gap-Init achieves a higher ceiling than LoRA under rank-1 adaptation.

Moreover, under extremely low-rank constraints, Gap-Init matches or surpasses stronger PEFT baselines such as DoRA, indicating that appropriate initialization can be more critical than increasing adaptation rank under identical experimental conditions. In particular, while the performance gap narrows with extended training, explicit alignment with the gap direction continues to provide a stronger initialization, accelerating convergence and improving performance.

Together, these results provide strong evidence that the proposed initialization-based adaptation strategy enables effective, robust, and parameter-efficient low-rank adaptation across models and training regimes.

Comprehensive ablation studies on calibration robustness, noise sensitivity, and cross-backbone generalization are presented in Appendices D, E, and F.

## 7. Conclusion

This work identifies a fundamental geometric bottleneck underlying the failure of rank-1 parameter-efficient fine-tuning in multimodal models. Through an analysis of representation geometry, we show that vision and text features form anisotropic cones whose discrepancy admits a translation-dominant component that disproportionately affects optimization under rank-1 constraints. This structure implies that standard random initialization is likely to misalign the rank-1 update direction with this component, leading to attenuated gradient flow and unstable optimization.

We introduce **Gap-Init**, a geometry-aware initialization strategy that aligns the LoRA update subspace with an estimated modality-gap direction in a simple effective manner. Gap-Init requires only a small domain-aligned calibration set and no additional parameters, yet consistently stabilizes rank-1 training and matches or slightly exceeds strong rank-8 baselines across captioning, VQA, zero-shot transfer, and hallucination robustness. Together, our analysis and experiments suggest that, at the extreme low-rank limit, direction can matter as much as rank. While our focus is on rank 1

adaptation, the analysis highlights a broader principle for extremely constrained adaptation regimes.

Unlike weight-space refinements such as DoRA or PiSSA, Gap-Init operates entirely in activation space and is designed for the pathological rank-1 regime, where optimization is direction-limited rather than capacity-limited. These findings highlight the importance of data- and activation-space geometry in PEFT and suggest that understanding representation alignment may unlock more efficient adaptation strategies for multimodal LLMs.

Gap-Init should not be interpreted as a universal improvement for all LoRA configurations. Its main role is to characterize and mitigate a direction-limited regime that becomes most visible under extreme low-rank multimodal adaptation. As rank, model scale, or training budget increases, the effect of initialization direction may diminish because the optimizer has greater capacity to discover useful adaptation directions.

At rank 1, the bottleneck shifts from rank to direction.

## Impact Statement

This work studies the geometric properties of multimodal representations and proposes a more stable and parameter-efficient approach for adapting large vision–language models. As our method reduces the computational cost of fine-tuning, it may broaden access to multimodal adaptation techniques for research groups with limited resources. More efficient fine-tuning may also reduce the energy footprint associated with training large models.

At the same time, scaling and adapting multimodal models raise important ethical considerations. Improved efficiency does not eliminate the risks inherent to pretrained models, such as biased associations, hallucination, or incorrect reasoning under distribution shift. Gap-Init focuses on the initialization of low-rank adapters and does not directly address these broader safety challenges. Care should therefore be taken when deploying adapted models in real-world settings, especially in environments involving high-stakes decisions or sensitive content.

Our analysis highlights the geometric structure of modality alignment but does not change the underlying data on which models were trained. As a result, any societal biases present in training data may persist after adaptation. We encourage practitioners to apply appropriate auditing methods and to evaluate models for fairness, robustness, and potential downstream harm.

Overall, we hope that this work contributes to a deeper understanding of multimodal representation learning, while also emphasizing the need for responsible evaluation and deployment of adapted large-scale models.

## Acknowledgements

This research was supported by the Institute of Information & communications Technology Planning & Evaluation (IITP) grant funded by the Korea government (MSIT) (No.RS-2026-25519206, Development of human-like intelligent generative agents based on interactive multimodal reverse prompting), and IITP grant funded by the Korea government(MSIT) (No.RS-2025-02217259, Development of self-evolving AI bias detection-correction-explain platform based on international multidisciplinary governance), (RS-2024-00395401, Development of VFX creation and combination using generative AI).

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

# A. Gaussian Toy Model for Rank-1 Optimization Collapse

This appendix provides a formal analysis supporting the geometric interpretation of rank-1 optimization collapse discussed in the main text. We introduce a minimal Gaussian translation model and show that, in high dimensions, randomly initialized rank-1 LoRA updates suffer severe gradient suppression, while alignment-aware initialization avoids this issue.

### A.1. Gaussian translation model.

We model visual and textual hidden states as two isotropic Gaussian clusters in $\mathbb{R}^d$:

$$x_v \sim \mathcal{N}(\mu_v, \sigma^2 I_d), \qquad x_t \sim \mathcal{N}(\mu_v + \vec{g}, \sigma^2 I_d), \tag{2}$$

where $\vec{g}$ denotes the modality-gap translation vector. Due to spherical covariance, $\vec{g}$ is the only non-isotropic structure in the space.

### A.2. Rank-1 Update and Optimal Direction

A rank-1 LoRA adapter $\Delta W = ba^\top$ induces the update $x_v \mapsto x_v + \Delta W x_v$, leading to the expected alignment loss

$$\mathcal{L} = \mathbb{E}\|x_v + \Delta W x_v - x_t\|^2. \tag{3}$$

Taking expectations over the Gaussian distributions reduces the optimization problem to

$$\min_{b,\theta} \|\theta b - \vec{g}\|^2,$$

which implies that the optimal direction satisfies

$$b^* \parallel \vec{g}. \tag{4}$$

This shows that any successful rank-1 update must recover the modality-gap direction.

### A.3. Gradient suppression under random initialization.

Standard LoRA initializes $b$ uniformly at random on the unit sphere: $b \sim \text{Unif}(\mathbb{S}^{d-1})$. The useful gradient component is proportional to its projection onto $\vec{g}$:

$$\nabla \mathcal{L} \propto \langle b, \vec{g} \rangle. \tag{5}$$

For a random direction in $\mathbb{R}^d$, the squared cosine similarity $\langle b, \hat{g} \rangle^2$ follows a $\text{Beta}(\frac{1}{2}, \frac{d-1}{2})$ distribution, yielding

$$\mathbb{E}[\langle b, \hat{g} \rangle^2] = \frac{1}{d}, \qquad |\langle b, \hat{g} \rangle| = \mathcal{O}(d^{-1/2}). \tag{6}$$

### A.4. Optimization Collapse Theorem

**Theorem A.1** (Optimization Collapse). *Under the Gaussian translation model, a randomly initialized rank-1 LoRA update experiences $\mathcal{O}(d^{-1/2})$ suppression of the useful gradient component $\langle b, \vec{g} \rangle$. As dimensionality increases, this places the optimizer in a flat region where the modality-gap direction cannot be reliably recovered. Initializing $b \propto \vec{g}$ restores $\mathcal{O}(1)$ gradient flow and avoids this collapse.*

### A.5. Synthetic validation.

We validate the predicted scaling behavior using synthetic Gaussian data. Random rank-1 directions exhibit the expected $\mathcal{O}(d^{-1/2})$ decay, while gap-aligned initialization maintains a constant signal across dimensions (Fig. 5). This confirms that the observed rank-1 collapse arises from high-dimensional geometry rather than insufficient representational capacity.

# B. Why Translation Dominance Emerges in Multimodal Alignment

This appendix provides a mechanistic discussion supporting the empirical observation that the modality gap in modern vision–language models is often dominated by a translation-like component. While multimodal alignment is not intrinsically low-dimensional, several properties of contemporary training objectives and architectures bias the misalignment toward a coherent global shift.

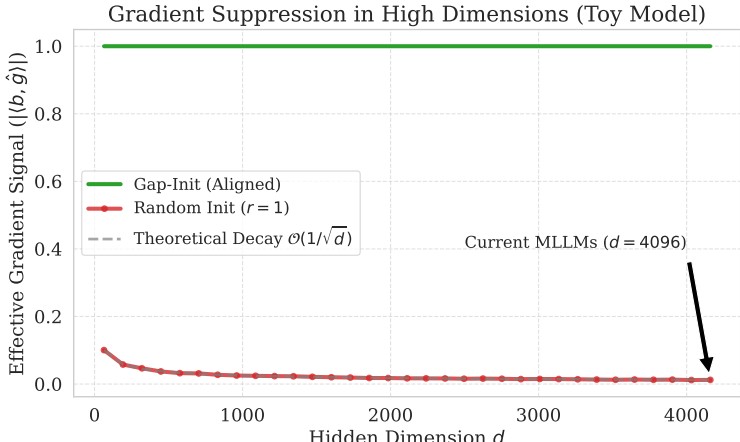

*Figure 5.* **Gradient suppression in high dimensions (Gaussian toy model).** We compute the effective gradient magnitude $|\langle b, \hat{g} \rangle|$ for Rank-1 LoRA directions in dimensions $d \in [64, 4096]$. Random initialization exhibits the predicted $\mathcal{O}(d^{-1/2})$ decay (red), showing that almost all gradient signal along the modality gap vanishes at high dimensionality. In contrast, GAP-INIT (green) maintains $\mathcal{O}(1)$ signal independent of dimensionality. This verifies that optimization collapse in Rank-1 LoRA is a consequence of high-dimensional geometry, not insufficient rank.

### B.1. Contrastive objectives encourage cone formation.

Contrastive pretraining objectives (e.g., CLIP, BLIP-2) optimize representations by pulling matched pairs together while pushing unmatched pairs apart. This process induces strong global anisotropy in the embedding space (Mu & Viswanath, 2018b; Ethayarajh, 2019), causing representations to collapse into modality-specific cones. Once such a cone structure emerges, the dominant difference between two modalities is well-approximated by the displacement between their centroids, naturally giving rise to a translation-like gap.

### B.2. Heterogeneous encoder architectures introduce systematic bias.

Vision encoders and language models are architecturally distinct and trained under different normalization and scaling conventions. These systematic differences (e.g., residual-path statistics, tokenization effects) manifest as coherent shifts in embedding means. Because these biases are consistent across instances, they accumulate into a stable, global translation vector rather than instance-specific distortions.

### B.3. Multimodal bridges operate approximately linearly.

Many vision–language models rely on shallow multimodal bridging modules (e.g., the Q-Former in BLIP-2) to project visual representations into the language embedding space. Relative to the frozen language model, these modules operate in a regime that is approximately linear. As a result, the dominant mismatch they must correct is also well captured by a linear, translation-like transformation.

### B.4. Downstream alignment requires only coarse repositioning.

Downstream tasks such as visual question answering and image captioning primarily require semantic compatibility between modalities, rather than fine-grained geometric warping of representations. Consequently, effective alignment often amounts to shifting visual features toward the linguistic manifold, without reshaping their internal structure. A translation captures this coarse, first-order correction.

### B.5. Summary

Taken together, strong embedding anisotropy, systematic architectural offsets, and approximately linear multimodal bridging jointly bias the modality gap toward a translation-dominant structure. This provides an intuitive explanation for why a rank-1 update aligned with the translation direction can exert a disproportionate effect on optimization, despite the underlying

alignment problem being higher-dimensional.

## C. Full Definition of the SPOT-Gap Metric

We provide the full formulation of the *SPOT-Gap* metric, used in Section 4.4 to quantify layer-wise modality misalignment. SPOT-Gap stands for **Semantic Preserving Optimal Transport–Gap**. It is a lightweight, training-free diagnostic inspired by optimal-transport principles: instead of solving a full OT plan, we estimate the semantic displacement required to transport visual features onto the textual manifold while preserving local structure. SPOT-Gap consists of three components: (1) sample-projected semantic gap, (2) structure-preserving geometric discrepancy, and (3) neighborhood-consistency smoothing.

Let $H_v^{(l)} = \{h_{v,i}^{(l)}\}$ and $H_t^{(l)} = \{h_{t,i}^{(l)}\}$ denote visual and textual hidden states at layer $l$, extracted from a calibration set $\mathcal{D}_{\text{cal}}$. All vectors lie in $\mathbb{R}^d$.

### C.1. Sample-Projected Semantic Gap

For each paired sample $(i)$, we consider the semantic displacement needed to align the visual feature to its textual counterpart:

$$g_{\text{sem},i}^{(l)} = h_{t,i}^{(l)} - h_{v,i}^{(l)}.$$

The magnitude

$$s_{\text{sem},i}^{(l)} = \|g_{\text{sem},i}^{(l)}\|_2$$

measures the semantic transport cost of matching the two modalities at the sample level. This is a first-order approximation of the cost of transporting $h_{v,i}^{(l)}$ toward $h_{t,i}^{(l)}$ under a diagonal OT coupling.

### C.2. Structure-Preserving Geometric Discrepancy

To preserve local semantic structure—a key OT concept—we incorporate a discrepancy term measuring how local geometry differs between modalities. For each feature $h_{v,i}^{(l)}$, let $\mathcal{N}_k(h_{v,i}^{(l)})$ denote its $k$ nearest neighbors in $H_v^{(l)}$. We compute its local dispersion:

$$d_{v,i}^{(l)} = \frac{1}{k} \sum_{j \in \mathcal{N}_k(h_{v,i}^{(l)})} \|h_{v,i}^{(l)} - h_{v,j}^{(l)}\|_2.$$

Similarly, compute $d_{t,i}^{(l)}$ from $H_t^{(l)}$.

The geometric discrepancy is defined as

$$s_{\text{geom},i}^{(l)} = \left| d_{t,i}^{(l)} - d_{v,i}^{(l)} \right|.$$

This captures how much structural distortion is needed to locally "transport" visual neighborhoods into textual neighborhoods.

### C.3. Neighborhood-Consistency Smoothing

To reduce noise and strengthen structural preservation, we smooth the combined semantic and geometric scores over each sample's neighborhood:

$$\bar{s}_i^{(l)} = \frac{1}{|\mathcal{N}_k(i)|} \sum_{j \in \mathcal{N}_k(i)} \left( \alpha_{\text{sem}} s_{\text{sem},j}^{(l)} + \alpha_{\text{geom}} s_{\text{geom},j}^{(l)} \right),$$

where $\alpha_{\text{sem}}$ and $\alpha_{\text{geom}}$ are fixed weights. This step approximates OT's global smoothness constraints without solving a full transport map.

### C.4. Final SPOT-Gap Score

The layer-wise SPOT-Gap score is the mean smoothed discrepancy:

$$S_{\text{gap}}^{(l)} = \frac{1}{N} \sum_{i=1}^{N} \bar{s}_i^{(l)}.$$

---

**Algorithm 2** SPOT-Gap Scoring Procedure (Layer $l$)

---

1: **Input:** Visual hidden states $H_v^{(l)} = \{h_{v,i}^{(l)}\}$, textual hidden states $H_t^{(l)} = \{h_{t,i}^{(l)}\}$, neighborhood size $k$, weights $\alpha_{\text{sem}}, \alpha_{\text{geom}}$.
2: **Output:** SPOT-Gap score $S_{\text{gap}}^{(l)}$.
3: **Step 1: Compute sample-projected semantic gaps**
4: **for** each sample $i$ **do**
5:     $g_{\text{sem},i} \leftarrow h_{t,i}^{(l)} - h_{v,i}^{(l)}$
6:     $s_{\text{sem},i} \leftarrow \|g_{\text{sem},i}\|_2$
7: **end for**
8: **Step 2: Compute geometric discrepancy**
9: **for** each sample $i$ **do**
10:     Identify $k$ nearest neighbors $\mathcal{N}_k(h_{v,i}^{(l)})$ in $H_v^{(l)}$
11:     $d_{v,i} \leftarrow \frac{1}{k} \sum\limits_{j \in \mathcal{N}_k(h_{v,i}^{(l)})} \|h_{v,i}^{(l)} - h_{v,j}^{(l)}\|_2$
12:     Identify $k$ nearest neighbors $\mathcal{N}_k(h_{t,i}^{(l)})$ in $H_t^{(l)}$
13:     $d_{t,i} \leftarrow \frac{1}{k} \sum\limits_{j \in \mathcal{N}_k(h_{t,i}^{(l)})} \|h_{t,i}^{(l)} - h_{t,j}^{(l)}\|_2$
14:     $s_{\text{geom},i} \leftarrow |d_{t,i} - d_{v,i}|$
15: **end for**
16: **Step 3: Neighborhood-consistency smoothing**
17: **for** each sample $i$ **do**
18:     Identify $k$ nearest neighbors $\mathcal{N}_k(i)$ across paired samples
19:     $\bar{s}_i \leftarrow \frac{1}{|\mathcal{N}_k(i)|} \sum\limits_{j \in \mathcal{N}_k(i)} (\alpha_{\text{sem}}\, s_{\text{sem},j} + \alpha_{\text{geom}}\, s_{\text{geom},j})$
20: **end for**
21: **Step 4: Final layer score**
22: $S_{\text{gap}}^{(l)} \leftarrow \frac{1}{N} \sum_{i=1}^{N} \bar{s}_i$
23: **return** $S_{\text{gap}}^{(l)}$

---

**Interpretation.** A high $S_{\text{gap}}^{(l)}$ indicates that layer $l$ exhibits large semantic displacement and structural incongruence between modalities—precisely the characteristics that make a layer difficult to align. Thus, SPOT-Gap acts as a semantic-preserving, OT-inspired diagnostic for identifying layers with severe multimodal misalignment.

**Usage.** In Section 4.4, we use SPOT-Gap to validate our geometric analysis and to guide optional layer selection. While Gap-Init performs strongly even under naive selection, SPOT-Gap allows us to probe internal modality structure and confirm alignment-sensitive layers.

## D. Calibration Set Size and Data Composition

### D.1. Calibration Set Size Sensitivity

To evaluate the robustness of the estimated modality-gap vector $g$, we vary the calibration set size $|\mathcal{D}_{\text{cal}}| \in \{16, 64, 256, 1024\}$ and report results on COCO Captioning using CIDEr and BLEU-4.

As shown in Table 9, the effectiveness of Gap-Init depends on obtaining a statistically representative estimate of the modality gap. With extremely small calibration sets ($|\mathcal{D}_{\text{cal}}| = 16$), performance degrades due to high variance in the estimated direction, suggesting insufficient averaging of instance-specific noise.

Performance improves rapidly as the calibration size increases, with a clear peak at $|\mathcal{D}_{\text{cal}}| = 256$. Further increasing the size to 1024 yields diminishing returns, indicating that the modality gap corresponds to a low-rank, global geometric feature that does not require large-scale sampling. Unless otherwise specified, we adopt $|\mathcal{D}_{\text{cal}}| = 256$ as the default configuration.

*Table 9.* **Effect of calibration set size $|\mathcal{D}_{\text{cal}}|$ on rank-1 adaptation.** Performance peaks at a moderate calibration size, indicating that the global translation direction can be robustly estimated without exhaustive sampling.

| $|D_{\text{cal}}|$ | CIDEr | BLEU-4 |
|---|---|---|
| 16 | 109.99 | 32.36 |
| 64 | 137.68 | 40.47 |
| 256 | **142.05** | **42.30** |
| 1024 | 140.03 | 41.36 |

*Table 10.* **Effect of calibration data source on rank-1 adaptation.** Pure out-of-distribution calibration degrades performance, while mixing compatible domains yields a slight improvement over the in-domain baseline.

| Calibration Source | CIDEr | BLEU-4 |
|---|---|---|
| Flickr30k (OOD) | 108.85 | 31.50 |
| COCO (In-Domain) | 140.56 | 41.44 |
| Mixed (50% COCO + 50% Flickr) | **141.91** | **41.95** |

### D.2. Calibration Data Composition: Domain Specificity vs. Diversity

While the previous subsection studies calibration size, we next examine the effect of calibration data distribution. We fix $|\mathcal{D}_{\text{cal}}| = 256$ and evaluate Gap-Init using: (i) pure in-domain data (COCO), (ii) pure out-of-distribution data (Flickr30k), and (iii) a mixed calibration set (50% COCO, 50% Flickr30k).

Results in Table 10 highlight two key properties of the modality-gap estimation.

First, the modality gap exhibits domain specificity. Calibration using purely out-of-distribution data (Flickr30k) leads to a substantial performance drop compared to in-domain calibration, indicating that the translation direction varies across data distributions.

Second, introducing compatible diversity can act as a mild regularizer. The mixed calibration set slightly outperforms the pure in-domain baseline. Since Flickr30k and COCO share a similar semantic manifold of natural images, mixing the two does not disrupt the principal translation direction but introduces additional variability that stabilizes the estimation. This suggests that effective rank-1 initialization lies in a region that is well aligned with the target domain while remaining sufficiently broad to improve robustness.

## E. Robustness to Noise in the Gap Direction

### E.1. Noise Injection into the Gap Vector

To assess the sensitivity of Gap-Init to directional perturbations, we construct noisy gap vectors

$$g' = \text{normalize}(g + \epsilon \cdot \eta),$$

where $\eta \sim \mathcal{N}(0, I)$ and $\epsilon \in \{0.0, 0.2, 0.6, 1.0\}$. Results are reported in Table 11.

As shown in Table 11, performance exhibits a non-monotonic response to noise injection. At low noise levels ($\epsilon = 0.2$), performance decreases slightly, suggesting that the estimated gap direction is already locally well aligned and small perturbations can be detrimental.

At a moderate noise level ($\epsilon = 0.6$), performance recovers and slightly improves, with BLEU-4 reaching its peak. This behavior is consistent with a stochastic regularization effect, where initializing slightly off-center from the estimated direction encourages exploration of a broader basin of attraction.

In contrast, when noise dominates the signal ($\epsilon = 1.0$), performance degrades sharply. This regime serves as a negative control: once directional information is largely destroyed, rank-1 adaptation becomes ineffective, confirming that Gap-Init depends critically on preserving the semantic alignment encoded in the translation direction.

*Table 11.* **Effect of Noise Injection** ($\epsilon$). Metrics in percentage. While the method is sensitive to large deviations ($\epsilon = 1.0$), moderate noise ($\epsilon = 0.6$) yields the highest BLEU-4 score.

| Noise Level ($\epsilon$) | CIDEr | BLEU-4 |
|---|---|---|
| 0.0 | 141.84 | 42.55 |
| 0.2 | 140.04 | 41.62 |
| 0.6 | **142.00** | **43.43** |
| 1.0 | 133.45 | 39.62 |

*Table 12.* Cross-backbone generalization of Gap-Init under low-rank adaptation. Metrics are reported in percentage. Gemma3-4B results are shown under both single-epoch and five-epoch training budgets.

| Method | CIDEr | BLEU-4 | METEOR |
|---|---|---|---|
| **Qwen2-VL-7B-Instruct** | | | |
| Rank=8, Random | 142.57 | 41.72 | 32.04 |
| Rank=1, Random | 143.67 | 42.55 | 31.89 |
| Rank=1, Gap-Init | **144.11** | **42.97** | **31.95** |
| **Gemma3-4B (1 epoch)** | | | |
| Rank=8, Random | 89.62 | 22.67 | 23.86 |
| Rank=1, Random | 90.35 | 24.95 | 24.28 |
| Rank=1, Gap-Init | **95.24** | **27.27** | **26.33** |
| **Gemma3-4B (5 epochs)** | | | |
| Rank=8, Random | 94.82 | 27.35 | 27.91 |
| Rank=1, Random | 95.43 | 28.12 | 28.36 |
| Rank=1, Gap-Init | **96.17** | **28.89** | **28.74** |

# F. Additional Cross-Backbone Results

This appendix provides additional results on the generalization of Gap-Init across vision–language backbones with different architectures, scales, and training budgets. These experiments complement the main results by illustrating how explicit alignment with the modality-gap direction behaves under varying model capacity and optimization regimes.

## F.1. Cross-Model Generalization Across Vision–Language Backbones

We evaluate Gap-Init on two representative backbones: Qwen2-VL-7B-Instruct, a large-scale vision–language model with substantial capacity, and Gemma3-4B, a smaller model operating under more constrained optimization conditions. Results are reported in Table 12.

On the large-scale backbone Qwen2-VL-7B, rank-1 adaptation with random initialization already performs competitively with higher-rank baselines, indicating reduced sensitivity to rank when model capacity is sufficient. Nevertheless, applying Gap-Init at rank 1 consistently improves performance across all evaluation metrics, yielding the best overall results among the compared settings.

On the smaller-scale backbone Gemma3-4B, the benefits of Gap-Init are more pronounced. Under a single-epoch training budget, rank-1 adaptation with random initialization provides only modest gains, whereas Gap-Init leads to substantial improvements across all metrics, including a significant increase in CIDEr score. This highlights the importance of aligning low-rank updates with dominant cross-modal discrepancies when optimization resources are limited.

Importantly, this advantage persists as the training budget increases. With five epochs of training, randomly initialized low-rank updates partially recover effective directions, reducing the performance gap. However, Gap-Init continues to outperform both rank-1 and rank-8 random baselines, indicating that explicit alignment with the modality-gap direction provides a stronger initialization that accelerates convergence and improves final performance.

*Table 13.* LoRA configuration details.

| Parameter | Value |
|---|---|
| Rank ($r$) | 1 (main), 8 (baseline comparison) |
| Scaling factor ($\alpha$) | 2 (rank 1), 16 (rank 8) |
| LoRA dropout | 0.05 |
| Initialization (baseline) | Random Gaussian |
| Initialization (Gap-Init) | $B^{(\ell)} \leftarrow g^{(\ell)}/\|g^{(\ell)}\|_2, A^{(\ell)} \leftarrow 0$ |

*Table 14.* Training hyperparameters.

| Parameter | Value |
|---|---|
| Optimizer | AdamW |
| Learning rate | $2 \times 10^{-4}$ |
| Learning rate scheduler | Cosine decay |
| Batch size (per device) | 32 |
| Gradient accumulation steps | 2 |
| Effective batch size | 64 |
| Training epochs | 5 |
| Precision | FP16 / BF16 (depending on hardware) |
| Random seeds | $\{42, 123, 2024, 999, 7\}$ |

# G. Implementation Details

This appendix provides complete implementation details to ensure reproducibility of all experiments involving Gap-Init and related baselines.

## G.1. Model Architecture and Adaptation Scope

Unless otherwise specified, we conduct experiments on BLIP-2 with an OPT-2.7B language model backbone. The hidden size of the language model is $d = 2560$. Visual features are aligned into the language model space via a frozen Q-Former followed by a linear projection layer. All modality gap computations and LoRA adaptations are performed in the language model hidden-state space after this alignment.

**Adapted modules.** LoRA adapters are inserted into linear layers of both the language model and (optionally) the vision encoder. Specifically, we target the following submodules:

- Attention layers: `q_proj, k_proj, v_proj, o_proj`

- MLP layers: `fc1, fc2`

The Q-Former is frozen in all Gap-Init experiments unless stated otherwise.

## G.2. LoRA Configuration

Table 13 summarizes the LoRA hyperparameters used throughout the paper.

For rank-$r > 1$, only the first column of $B^{(\ell)}$ is aligned with the modality gap; remaining columns are randomly initialized.

## G.3. Training Hyperparameters

Table 14 reports the training hyperparameters used for captioning and VQA tasks.

All experiments are conducted with identical training schedules across initialization methods to ensure fair comparison.

### G.4. Calibration Protocol for Modality Gap Estimation

The modality gap vectors are estimated using a small, unlabeled calibration set $\mathcal{D}_{\text{cal}}$. Unless otherwise specified, calibration samples are drawn from the COCO Train2017 split (in-domain).

**Calibration size.** We use 256 image–text pairs, which provides a stable estimate of the gap while incurring negligible computational overhead. Ablations on calibration size are reported in Section 5.

**Procedure.** The calibration process proceeds as follows:

1. Randomly sample $(x_{\text{img}}, x_{\text{txt}})$ pairs from $\mathcal{D}_{\text{cal}}$.

2. Perform a forward pass using the frozen pretrained model.

3. For each transformer layer $\ell$, extract:
   - aligned visual hidden states $h_v^{(\ell)}$,
   - textual hidden states $h_t^{(\ell)}$.

4. Compute the sample-projected gap $g^{(i,\ell)} = h_t^{(\ell)} - h_v^{(\ell)}$.

5. Apply SPOT-Gap smoothing (Appendix C) and average over samples to obtain $g^{(\ell)}$.

The resulting gap vectors are stored and reused for initialization; no gradients are computed during this stage.

### G.5. Layer Selection Strategies

We consider two layer selection strategies:

- **Naive selection:** the top-$k$ transformer layers.

- **Gap-guided selection (GG-Safe):** the top-$k$ layers ranked by the SPOT-Gap score, excluding early vision encoder layers for safety.

Unless otherwise noted, Gap-Init is applied using naive selection; GG-Safe is used only to validate the diagnostic value of the modality gap.

### G.6. Notes on Full Fine-Tuning

We do *not* perform full fine-tuning of the backbone model in this work. Any reference to a "full fine-tuning limit" in figures denotes an approximate upper bound derived from prior reported results and is not an experimental baseline trained by us. This will be clarified explicitly in the revised manuscript.

