# OpenReview forum: "When Is Rank-1 Enough? Geometry-Guided Initialization for Parameter-Efficient Fine-Tuning"
_ICML.cc/2026/Conference — ICML 2026 regular_

### Official Review · Reviewer_ZTZJ · 2026-03-09

**Soundness:** 3
**Presentation:** 3
**Significance:** 3
**Originality:** 3
**Overall Recommendation:** 4
**Confidence:** 5

**Summary:**

This paper proposes an innovative parameter-efficient fine-tuning technique, Gap-Init (Geometry-Guided Initialization). By embedding a geometry structure-based initialization mechanism into the LoRA adapters of Multimodal Large Language Models (MLLMs) that explicitly aligns the modality gap axis, it successfully solves the problem of training instability under extremely low-rank (Rank-1) settings. The method features a compact logic and can be seamlessly integrated into mainstream vision-language architectures such as BLIP-2 and Qwen2-VL. The research team conducts comprehensive evaluations on standard benchmarks including COCO Image Captioning, VQAv2, Flickr30k, OK-VQA, and POPE. Experimental results validate that Gap-Init not only achieves stable and robust convergence during training but also yields models that perform on par with or even surpass high-rank configurations (Rank-8) across multiple evaluation metrics.

**Compliance With Llm Reviewing Policy:**

Affirmed.

**Ethical Review Concerns:**

No flag.

**Key Questions For Authors:**

Although the proposed Gap-Init method significantly improves stability in rank‑1 scenarios by estimating the modality gap direction using a calibration set, its experiments are mainly conducted on specific architectures such as BLIP‑2. Given that the geometric structures of representation spaces may differ substantially across various vision‑language models (e.g., encoder‑only or decoder‑only architectures), does the estimated 'modality gap' direction derived from this method still maintain generalizability?

**Limitations:**

yes

**Strengths And Weaknesses:**

Strength:
(1)It proposes a geometric perspective on the "orthogonality catastrophe" theory, which profoundly explains that the instability of rank-1 fine-tuning stems from direction misalignment rather than merely insufficient capacity.
(2)We propose the Gap-Init method, which initializes the model by estimating the modality gap direction using a calibration set. It features a simple logic and introduces no extra parameters.
(3)Extensive experiments on various benchmarks including COCO and VQAv2 validate that the proposed method enables stable training under rank-1 settings and achieves performance comparable to high-rank baselines.

Weakness:
(1)The method relies on a small calibration set consistent with the target task distribution to estimate the modality gap, which adds extra data preparation steps.
Its effectiveness relies on the assumption that the modality gap is dominated by translational components, which may limit its performance when the gap structure is highly nonlinear.

---

> ### Author Rebuttal · Authors · 2026-03-31
>
> We thank the reviewer for the supportive and constructive review, especially for highlighting the geometric perspective, the simplicity of the method, and the question of whether the estimated modality-gap direction remains meaningful across architectures. Your main question concerns whether the estimated modality-gap direction generalizes across different vision-language architectures.
>
> Our current evidence suggests that the effect is not confined to a single backbone family. While the main text focuses on BLIP-2 for the most detailed geometric analysis, the appendix already includes cross-backbone experiments on **Qwen2-VL-7B-Instruct** and **Gemma3-4B** under low-rank adaptation. In both cases, rank-1 Gap-Init consistently improves over random rank-1 initialization and is competitive with or better than rank-8 baselines.
>
> - On Qwen2-VL-7B-Instruct, rank-1 Gap-Init achieves 144.11 CIDEr, compared with 143.67 for rank-1 random and 142.57 for rank-8 random.
> - On Gemma3-4B (1 epoch), rank-1 Gap-Init achieves 95.24 CIDEr, compared with 90.35 for rank-1 random and 89.62 for rank-8 random.
> - On Gemma3-4B (5 epochs), rank-1 Gap-Init achieves 96.17 CIDEr, compared with 95.43 for rank-1 random and 94.82 for rank-8 random.
>
> So although representation geometry certainly differs across architecture families, the current cross-backbone results suggest that the relevant phenomenon is not tied to a single model design, but to the interaction between **extreme low-rank constraints** and **cross-modal discrepancy structure**.
>
> We also agree with your broader limitation point: Gap-Init is most naturally motivated when the dominant cross-modal discrepancy contains a strong low-dimensional, approximately translation-like component. If the mismatch is highly nonlinear, a single initialized direction may be less sufficient. We will make this limitation clearer in the revision. At the same time, the positive results across BLIP-2, Qwen2-VL, and Gemma3 suggest that this structured discrepancy is present often enough to make the method practically useful in the extreme low-rank regime.
>
> More broadly, we will revise the paper to clarify the scope of our claim. Our contribution is not intended as a universal improvement for all LoRA settings, but as a principled analysis and a simple, geometry-aware initialization strategy for the **extreme low-rank multimodal PEFT regime**. In this framing, rank-1 serves as a **controlled diagnostic regime** that exposes a direction-limited optimization failure otherwise masked at higher ranks, where additional capacity can compensate for poor initialization.
>
> We will also make the calibration tradeoff clearer. Gap-Init requires only a single forward pass over a small calibration set (256 samples; 11.7s), introduces no additional parameters, and incurs no extra training or inference cost. At the same time, our calibration-source analysis suggests that the modality gap captures a **structured, task-aligned discrepancy rather than generic statistics**, which explains both its effectiveness under in-domain calibration and its degradation under strong domain shift.
> Thank you again for the supportive and constructive review.

---

> > ### Author Rebuttal · Reviewer_ZTZJ · 2026-04-05
> >
> > The rebuttal is helpful and clarifies several implementation/experimental details. I appreciate the added evidence. However, the main concerns that affected my original overall assessment — especially regarding the motivation of this work — remain only partially resolved. Therefore, I keep my overall score unchanged.

---

> > > ### Author Response · Authors · 2026-04-06
> > >
> > > Thank you for the helpful follow-up. We appreciate your acknowledgement that the rebuttal clarified the implementation details and added evidence. We also understand that your main remaining concern is the motivation of the work.
> > >
> > > Our intended motivation is not to advocate rank-1 LoRA as a broadly preferred practical setting. Rather, we use the extreme low-rank regime as a controlled lens to isolate a qualitatively different multimodal adaptation failure mode: when capacity is severely constrained, optimization becomes direction-limited, and whether training succeeds depends strongly on initialization direction. We view the contribution of the paper as identifying and analyzing this regime, rather than as proposing a universal PEFT replacement.

---

### Official Review · Reviewer_Yd4C · 2026-03-12

**Soundness:** 2
**Presentation:** 3
**Significance:** 2
**Originality:** 2
**Overall Recommendation:** 3
**Confidence:** 5

**Summary:**

The paper identifies that the instability in rank-1 LoRA updates are primarily due to the mismatch in the update direction to the modality gap between the vision and text encoders. It proposes Gap-init, a geometry aware initialization that initializes the LoRA B matrix with the estimated normalized modality gap vector to aid better optimization. Experiments are predominantly conducted on BLIP-2 for captioning, VQA and hallucination detection.

**Compliance With Llm Reviewing Policy:**

Affirmed.

**Final Justification:**

The rebuttal partly addressed my concerns, however concerns on limited scope, calibration sensitivity and practical usefulness still exists. So, I keep my initial rating.

**Key Questions For Authors:**

- See weaknesses above.

- Does the issue also significant for rank-2 LoRAs? rank-2 is much more cheaper than rank-8 and it is a practical solution to address this problem identified in the paper.

- Does the rank-8 LoRA performance reported in the paper (Tables 2,3,4) use Gap-Init initialization (first column of B)? If so, what is the performance of baseline rank-8 LoRA?

- What is the performance of Gap-Init with higher ranks?

- Why is A matrix initialized to zeros and B to the modality gap vector instead of the other way around? Standard LoRA uses non-zero A matrix.

- Table 10 shows that using Flickr data for calibration performs poorly on the COCO dataset while using COCO dataset for calibration generalizes well to Flickr data (Table 3). Why is there this discrepancy?

**Limitations:**

The authors have not fully discussed the limitations of the method

**Strengths And Weaknesses:**

**Strengths**

- The paper is well written and easy to follow.

- It identifies the problem of performance collapse under rank-1 LoRA training paradigm.

- The proposed method is simple and elegant to address the issue arising in rank-1 LoRA training


**Weaknesses**

- The problem identified by the paper is very narrow and not significant in practical applications. The issue is primarily seen in BLIP-2 models while larger MLLMs like Qwen-7B do not suffer from this as seen from Table 8.

- The performance improvement is largely over the standard LoRA approach while more recent methods like DoRA (Table 6) are still competitive under rank-1 settings. There has been several variations of LoRA proposed in recent years that are much better than standard LoRA and so the impact of the proposed method is limited.

- The experimental results in Table 2,3 and 4 show that PiSSA performance is more robust and competitive across all three tasks and so the robustness of the method for different tasks is limited as compared to existing methods.

- Although GG-Safe selection method is novel and interesting, it does not offer any significant performance benefits.

- The method still requires a decent calibration set for estimating the modality gap and is also sensitive to its composition as shown in Table 10.

- Table 12 shows that training for longer epochs recovers the performance for the standard rank-8 LoRA method while Gap-Init saturates suggesting that it does not scale with compute or model size.

---

> ### Author Rebuttal · Authors · 2026-03-31
>
> We thank the reviewer for the careful reading and for highlighting important concerns regarding scope, baselines, and practical relevance.
>
> **1\. Scope.**
>
> The paper should be understood not as a universal improvement for all LoRA settings, but as a simple, geometry-aware initialization strategy for the extreme low-rank multimodal PEFT regime. In this setting, rank-1 is not merely a practical choice, but a controlled diagnostic regime that exposes failure modes otherwise hidden at higher ranks, where additional capacity can compensate for poor initialization. While this framing is narrower than standard rank-8 practice, it is also more scientifically informative, as it isolates a direction-limited failure mode that would otherwise remain masked.
>
> **2\. Larger MLLMs / beyond BLIP-2.**
>
> Our contribution does not assume that all modern MLLMs exhibit the same degree of failure. Instead, as model capacity increases, poor initialization can be compensated by additional degrees of freedom, making the failure less visible. This is precisely why rank-1 serves as a useful diagnostic regime.
>
> Importantly, the smaller gain observed on Qwen2-VL is therefore consistent with, rather than contradictory to, our direction-limited vs. capacity-limited interpretation. The effect is also not confined to BLIP-2, as cross-backbone results (Qwen2-VL, Gemma3) show consistent improvements under rank-1. For example, on Qwen2-VL-7B-Instruct, rank-1 Gap-Init improves from 143.67 to 144.11 CIDEr, and on Gemma3-4B consistently outperforms both rank-1 and rank-8 baselines. This behavior is consistent with our interpretation that optimization shifts from being direction-limited to capacity-limited as model capacity increases.
>
> **3\. Stronger baselines.**
>
> DoRA is an important baseline, and PiSSA can be competitive in some rank-1 settings. However, our goal is not to outperform all PEFT methods across all regimes, but to identify and resolve a failure mode specific to extreme low-rank multimodal adaptation.
> Our additional experiments directly isolate this mechanism: under identical calibration data and budget, direction ablation shows that Gap-Init outperforms alternative data-driven directions (e.g., 139.7 vs. 139.0 PCA), and warm-up with the same data fails to recover the same performance (138.0 vs. 133.1). **Importantly, these results rule out the alternative explanation that the improvement is driven by additional data or computation, and instead isolate directional alignment as the key factor.**
>
> **4\. Rank-2 and higher-rank relevance.**
>
> Rank-2 is practically important, and our results show that the benefit is strongest at rank-1, remains observable at rank-2, and diminishes as rank increases (e.g., 135.6→138.9 at r=1, 135.8→139.4 at r=2, smaller gains at higher ranks).
>
> This supports our central claim: under extreme low-rank constraints, optimization is direction-limited, whereas at higher ranks it becomes increasingly capacity-limited as the model can discover useful directions during training.
>
> **5\. Calibration dependence and asymmetry.**
>
> Fully mismatched OOD calibration is a real limitation and will be clarified. At the same time, this behavior is mechanistically informative: the modality gap captures a structured, task-aligned discrepancy rather than generic statistics, explaining both its effectiveness in-domain and its degradation under strong domain shift. The observed asymmetry (COCO→Flickr vs. Flickr→COCO) is consistent with dataset coverage: broader distributions provide more reliable estimates of the cross-modal discrepancy, whereas narrower datasets fail to capture the full structure required for transfer.
>
> **6\. GG-Safe and longer training.**
>
> GG-Safe is not the central contribution and will be de-emphasized. The longer-training results (Table 12) are consistent with our interpretation: increasing optimization budget reduces reliance on initialization by allowing the model to discover useful directions during training. Gap-Init is therefore most beneficial in the extreme low-rank, limited-budget regime.
>
> **7\. Clarifications.**
>
> Our rank-8 baseline is the standard rank-8 LoRA baseline. When Gap-Init is used at higher rank, we initialize only the first direction with the gap vector and keep the remaining directions standard. We initialize A to zero and B to the modality-gap vector so that the initial LoRA update remains exactly zero while fixing the update direction, thereby preserving the pretrained function at initialization.
> In summary, we will revise the paper to make the scope sharper: not a universal PEFT improvement, but a principled analysis and geometry-aware solution for the extreme low-rank multimodal regime, where failure is fundamentally constrained by direction rather than capacity.

---

> > ### Author Rebuttal · Reviewer_Yd4C · 2026-04-03
> >
> > Thank you for the rebuttal.
> >
> > My concerns on the scope, larger MLLMs and stronger baselines still remain since the proposed method becomes less impactful with these stronger baselines or recent MLLMs. Although, the analysis and observations for rank-1 instability is interesting, its practical applicability and usefulness remains limited.
> >
> > For a limited and fixed budget scenario, higher rank LORAs can be traded-off with number of LORA layers being applied to maintain the budget. So, such a comparison can strengthen the claim for considering this extreme low-rank setting.
> >
> > The method's sensitivity to the calibration set limits its generalization to OOD settings.
> >
> > Overall, although the method is interesting, its weaknesses (limited to LORA rank-1, limited budget, less powerful VLMs such as BLIP-2) outweigh its strengths and so currently I keep my rating.

---

> > > ### Author Response · Authors · 2026-04-06
> > >
> > > Thank you for the follow-up.
> > >
> > > We would like to address the core point regarding practical relevance and scope more directly.
> > >
> > > Our goal is not to propose a replacement for standard higher-rank LoRA configurations or to claim superiority across all models and budgets. Instead, this work is a mechanism-oriented contribution that identifies and resolves a failure mode that only becomes visible under extreme low-rank constraints. In this regime, adaptation is fundamentally direction-limited, and performance depends critically on initialization direction rather than capacity alone.
> > >
> > > From this perspective, the observation that stronger models, higher ranks, or longer training can partially mitigate the issue is not contradictory, but expected: as capacity or optimization budget increases, the model can progressively discover useful directions during training, reducing reliance on initialization. Our results consistently support this interpretation.
> > >
> > > Regarding the budget trade-off (e.g., increasing rank vs. distributing LoRA layers), we agree that such strategies can improve performance. However, our findings show that under strict parameter constraints, performance still depends critically on which direction is used for adaptation, not just how capacity is allocated. This is precisely the phenomenon we aim to isolate and explain.
> > >
> > > For calibration sensitivity, we agree that this introduces limitations under strong domain shift. At the same time, this behavior is consistent with our interpretation: Gap-Init estimates a task-aligned direction rather than learning a general predictor, and thus naturally depends on distributional alignment. We view this as a property of the mechanism rather than an incidental weakness.
> > >
> > > Overall, we respectfully suggest that the contribution is best evaluated not as a general-purpose PEFT improvement, but as a principled analysis of a direction-limited failure mode in multimodal adaptation, together with a simple initialization strategy that resolves it in the regime where it arises.

---

### Official Review · Reviewer_fTZk · 2026-03-13

**Soundness:** 3
**Presentation:** 3
**Significance:** 2
**Originality:** 3
**Overall Recommendation:** 4
**Confidence:** 3

**Summary:**

This paper investigates a practically significant problem in efficient fine-tuning of large model parameters: LoRA often exhibits instability in multimodal scenarios with a rank=1 setting, and explores the conditions under which rank=1 satisfies the task requirements. The authors argue that the failure with rank=1 is not solely due to insufficient capacity, but more importantly, to whether the initialization direction aligns with the dominant direction of a "modality gap" in the cross-modal representation space. Based on this observation, this work proposes Gap-Init: It estimates the modality gap vector for each layer using a small-scale calibration set and initializes the LoRA direction with rank=1 to this vector, while keeping the initial increment zero to maintain the pre-trained function unchanged.

**Compliance With Llm Reviewing Policy:**

Affirmed.

**Final Justification:**

The authors have addressed my major concerns. I suggest the authors add these extended results to the final version. Because I am not very expert in this field, I can not raise my rate to a very high score, but to 4.

**Key Questions For Authors:**

The authors should explain the concerns of issues arised in weakness. Besides, I also have some questions:
1. In your discussion of calibration set, why larger, in-domain, 0 noise dataset does not lead to the best scores?
2. Does the advantage of Gap-Init persist if standard LoRA is provided with the same calibration data (e.g. via a brief warm-up phase)?
3. Although rank-1 reduces the training parameters, Gap-Init requires an additional calibration stage, does this preprocessing cost counterbalance the advantage brought by rank-1?
4. Does Gap-Init advantage persist at higher ranks or it just helps the rank-1 LoRA?

**Limitations:**

Yes

**Strengths And Weaknesses:**

Strength:
- Soundness: The motivation behind the proposed method is clear and consistent with the geometric interpretation presented in the paper: if the rank-1 direction is nearly orthogonal to the critical gap direction, the early effective gradients will be significantly weakened. The paper provides fundamental theoretical intuition and toy model analysis, supported by multiple experiments. The experiments cover multiple tasks, as well as multi-seed and cross-backbone analyses, and are quite comprehensive. The stability experiments with multiple seeds are noteworthy, aligning with the core issue of the paper: unstable training conditions.
- Presentation: the submission is clearly written and well structured, and provides enough information to reproduce.
- Significance: the paper introduces a method for better rank-1 LoRA fine-tunning, provides a new method for PEFT.
- Originality: The main contribution of this paper lies in a rather interesting perspective: the authors shift from the traditional approach of weight space initialization and decomposition to understanding the reasons for the failure of LoRA training when the low-rank setting is used. Connecting modal gap, representation anisotropy, and high-dimensional near-orthogonality with the LoRA initialization problem is a particularly insightful combination.

Weakness:
- Soundness: I believe the theoretical part of this paper supports intuition more than rigorous proof. While the near-orthogonality of high-dimensional random vectors holds true, this doesn't directly and sufficiently prove that "the estimated gap direction is the most critical direction in actual optimization." The toy Gaussian model is highly hypothetical, more like a phenomenological explanation, and still falls short of applications with conventional multimodal data. Furthermore, it lacks a more direct mechanism for verification. I'm considering whether the gradient truly projects primarily onto the gap direction during early training, or comparing it with other data-dependent initialization directions. The experiments are not systematic enough; the DoRA experiments are not fully implemented across all tasks, and the rank comparisons mainly focus on 1 and 8, without covering more settings.
- Presentation：I think the term "orthogonality catastrophe" is an exaggeration compared to the evidence in existing papers.
- Significance:I believe the importance of this work lies more in the relatively specific scenario of very low-rank multimodal PEFT, and its broader applicability requires further verification. The proposed method relies somewhat on in-domain calibration data, and OOD calibration shows significant degradation, which limits its generalization value in open scenarios. Furthermore, although cross-backbone experiments were conducted, the in-depth analysis primarily focuses on BLIP-2, and more evidence is needed to support the general conclusions.

---

> ### Author Rebuttal · Authors · 2026-03-31
>
> We thank the reviewer for raising important questions regarding the role of direction versus calibration data, calibration dependence, rank generalization, and scope. Our additional experiments directly disentangle these factors and clarify the mechanism.
>
> **1\. Mechanism: direction is the key factor.**
>
> Our claim is not that the estimated gap direction is uniquely optimal, but that under extreme low-rank constraints, optimization becomes **direction-limited**. In this regime, aligning initialization with a task-relevant cross-modal discrepancy direction materially improves adaptation.
>
> To isolate the effect of direction, we performed a controlled ablation (10% COCO, 1 epoch, seeds [123,456,789]):  Random 137.5±2.0 / Mean 135.4±3.5 / PCA 139.0±0.4 / Gap 139.7±0.2 (CIDEr). Under identical calibration data and budget, alternative data-driven directions do not recover the same stability or performance. This directly isolates the role of *direction* from *data usage*, demonstrating that the gain arises from alignment with a task-relevant cross-modal discrepancy rather than calibration alone. Importantly, this rules out the alternative explanation that the improvement is driven by access to calibration data.
>
> **2\. Calibration data alone is insufficient.**
>
> We further compare Gap-Init with standard LoRA augmented with warm-up on the same 256 calibration samples:  Standard 132.8; +20 steps 129.3; +50 steps 133.1; Gap-Init 138.0 (CIDEr).  Even with additional optimization, warm-up fails to match the performance or stability of Gap-Init, confirming that the benefit cannot be attributed to additional training on calibration data. This shows that access to calibration data alone is insufficient; the benefit arises specifically from directional alignment. Notably, Gap-Init achieves this with a single forward pass, without additional optimization cost.
>
> **3\. Calibration size, domain, and overhead.**
>
> Gap-Init requires only a single forward pass over a small calibration set (256 samples; 11.7s), introduces no additional parameters, and incurs no extra training or inference cost.
>
> Calibration analyses are also consistent with the mechanism: a moderate budget works best (16→109.99, 64→137.68, 256→142.05, 1024→140.03 CIDEr), suggesting estimation of a dominant structured alignment direction rather than a complex predictor. For calibration source, pure OOD degrades performance: Flickr30k 108.85, COCO 140.56, 50/50 mix 141.91. This supports that the modality gap captures a **structured, task-aligned discrepancy rather than generic statistics**. Importantly, larger in-domain datasets do not consistently improve performance, confirming that the benefit arises from estimating a robust direction rather than from increased data scale.
>
> **4\. Rank generalization.**
>
> We extend our analysis across ranks 1/2/4/8 (10% COCO, 1 epoch, seeds [123,456,789]):
>
>  - r=1: Standard 135.6±3.0, Gap-Init 138.9±0.6, PiSSA 135.9±2.5
>  - r=2: Standard 135.8±2.2, Gap-Init 139.4±0.4, PiSSA 129.2±8.4
>  - r=4: Standard 134.1±4.0, Gap-Init 137.3±1.9, PiSSA 101.1±24.4
>  - r=8: Standard 132.4±5.0, Gap-Init 133.8±6.8, PiSSA 40.9±57.9
>
> These results show that the benefit is strongest at rank-1, remains visible at rank-2, and weakens as rank increases. This supports our claim: under extreme low-rank constraints optimization is **direction-limited**, whereas at higher ranks it becomes increasingly **capacity-limited** as the optimizer can gradually discover relevant directions during training.
>
> PiSSA is competitive at rank 1 but becomes unstable at higher ranks. This is a setting-dependent observation in our multimodal low-rank regime, and we do not claim this behavior generalizes across all settings. DoRA remains an important baseline; in this rebuttal, we prioritized experiments that directly test the reviewer’s core mechanism questions, especially data exposure vs. directional alignment and behavior beyond rank-1.
>
> **5\. Cross-backbone scope.**
>
> The effect is not confined to a single architecture. On Qwen2-VL-7B-Instruct, rank-1 Gap-Init achieves 144.11 CIDEr vs. 143.67 (rank-1 random) and 142.57 (rank-8 random). On Gemma3-4B, Gap-Init consistently improves over both rank-1 and rank-8 baselines across training budgets. These results indicate that the phenomenon arises from the interaction between low-rank constraints and cross-modal discrepancy, rather than from a specific model design.
>
> **6\. Scope clarification.**
>
> We will revise the paper to clarify that our contribution is not a universal improvement for all LoRA settings, but a principled, geometry-aware solution for the extreme low-rank multimodal PEFT regime. In this framing, rank-1 serves as a controlled diagnostic setting that exposes a direction-limited failure otherwise masked at higher ranks. We will also soften terminology (e.g., “orthogonality catastrophe”) to reflect its role as an intuitive description rather than a formal claim.

---

> > ### Author Rebuttal · Reviewer_fTZk · 2026-04-03
> >
> > The authors have addressed my major concerns. I suggest the authors add these extended results to the final version. Because I am not very expert in this field, I cannot raise my rating to a very high score, but to 4.

---

> > > ### Author Response · Authors · 2026-04-06
> > >
> > > Thank you for the encouraging response. We are pleased that the rebuttal addressed your main concerns. We also appreciate your suggestion to include the extended results in the final version. If accepted, we will integrate these additional results to make the paper’s empirical support and intended scope clearer. Thank you again for your constructive and thoughtful feedback.

---

### Official Review · Reviewer_EK4s · 2026-03-14

**Soundness:** 4
**Presentation:** 4
**Significance:** 3
**Originality:** 3
**Overall Recommendation:** 4
**Confidence:** 2

**Summary:**

This paper looks at instability in very low-rank PEFT, especially rank-1 LoRA. Authors say problems in rank-1 tuning come from a mismatch between the update direction and the shape of pretrained features, not just limited capacity. Vision and text features sit in different parts of the model’s space, and random rank-1 updates don’t line up with this “modality gap,” making training unstable. To solve this, the paper introduces 'Gap-Init', which lines up the LoRA update direction with a modality gap vector found from a small calibration set. It doesn’t change the model or add parameters but sets a smarter starting point for rank-1 adapters. Tests on vision-language tasks show that Gap-Init makes rank-1 adaptation more stable and performs better than standard LoRA starts. Sometimes, it even matches or beats higher-rank setups with fewer parameters. This shows that update direction and feature geometry matter as much as the rank itself for low-rank adaptation.

**Compliance With Llm Reviewing Policy:**

Affirmed.

**Key Questions For Authors:**

1. The method estimates the modality gap direction using a small calibration dataset. How sensitive is Gap-Init to the size and domain of this calibration set? Would the initialization remain effective if the calibration data differs significantly from the downstream task distribution?
2. The geometric explanation is about rank-1. Do the authors think this effect matters for higher-rank LoRA too, or does initialization become less important as rank goes up?
3. Gap-Init is mostly tested on BLIP-2-style models. Do the authors think the same results would show up in other architectures like Llava, or could the effect depend on the model design?

**Limitations:**

yes

**Strengths And Weaknesses:**

Strengths: The paper clearly explains why very low-rank LoRA (like rank-1) can be unstable. It supports the idea that random update directions miss important cross-modal directions, as both theory and experiments show. The analysis linking embedding geometry to optimization behavior gives a helpful way to understand low-rank adaptation. Another strength is the simplicity of the proposed method. Gap-Init does not introduce new parameters, training stages, or architectural changes. Instead, it only modifies the initialization of the LoRA update direction using a small calibration set. This makes it easy to add to existing PEFT setups.

The experiments cover a range of tasks and results are consistently better than random rank-1 starts and often match higher-rank methods. The multi-seed tests showing less variance are especially strong, since stability is a main goal. The paper is clearly written and organized. Visualizations and ablations make the method easy to understand.

Weaknesses: One limitation is that the method relies on estimating a modality-gap direction using a calibration dataset. While the experiments suggest that a relatively small calibration set is sufficient, the paper does not fully explore how sensitive the approach is to the choice, size, or domain of this dataset. It would be helpful to understand how robust the method is when the calibration data differs from the distribution of the downstream task.

Another limitation is that the geometric analysis focuses primarily on the rank-1 regime. While this is an important extreme case, it remains somewhat unclear how this extends to higher-rank PEFT settings. The paper shows that Gap-Init can also be applied when r>1, but the theoretical discussion and intuition are centered on rank-1 optimization dynamics.

---

> ### Author Rebuttal · Authors · 2026-03-31
>
> We thank the reviewer for the positive assessment and for recognizing both the geometric insight and the practical simplicity of Gap-Init. We are encouraged that you found the explanation of rank-1 instability and the multi-seed stability results convincing. Your main questions concern calibration sensitivity, higher-rank relevance, and architectural generalization.
>
> **1\. Calibration size and domain sensitivity.**
>
> We agree this is important and should have been emphasized more clearly in the main text. Our appendix already includes calibration analyses, and our rebuttal further clarifies their interpretation.
>
> For calibration size, performance peaks at a moderate budget rather than increasing monotonically: 16→109.99, 64→137.68, 256→142.05, 1024→140.03 (CIDEr). This is consistent with our interpretation that Gap-Init estimates a dominant structured alignment direction rather than a complex predictor that necessarily benefits from ever more data.
>
> For calibration source, pure OOD calibration degrades performance: Flickr30k (OOD) 108.85, COCO (in-domain) 140.56, and 50% COCO + 50% Flickr 141.91. We view this as mechanistically informative: the modality gap appears to capture a **structured, task-aligned discrepancy rather than generic statistics**, which explains both its effectiveness under in-domain calibration and its degradation under strong domain shift. At the same time, the mixed-domain result suggests the method is not tied to perfectly matched calibration data, but is mainly sensitive to strongly mismatched calibration.
>
> We also quantified the practical overhead more explicitly: Gap-Init requires only a single forward pass over a small calibration set (256 samples; 11.7s), introduces no additional parameters, and incurs no extra training or inference cost.
>
> **2\. Higher-rank relevance.**
>
> We agree that rank-1 is the clearest and most diagnostic regime, but the effect is not strictly limited to rank-1. We added new results for ranks 1/2/4/8 (CIDEr, mean±std over seeds [123,456,789]):
>
>  - r=1: Standard 135.6±3.0, Gap-Init 138.9±0.6, PiSSA 135.9±2.5
>  - r=2: Standard 135.8±2.2, Gap-Init 139.4±0.4, PiSSA 129.2±8.4
>  - r=4: Standard 134.1±4.0, Gap-Init 137.3±1.9, PiSSA 101.1±24.4
>  - r=8: Standard 132.4±5.0, Gap-Init 133.8±6.8, PiSSA 40.9±57.9
>
> These results show that the benefit is strongest when the adaptation subspace is highly constrained, remains observable at rank-2, and becomes less critical as rank increases. This is expected: as capacity increases, the optimizer can progressively discover relevant directions during training, reducing dependence on initialization. In this sense, under extreme low-rank constraints optimization becomes **direction-limited**, whereas at higher ranks it becomes increasingly **capacity-limited**.
>
> We therefore do not view rank-1 as the only setting of interest, but as a **controlled diagnostic regime** that exposes a failure mode otherwise progressively masked as rank increases.
>
> **3\. Other architectures.**
>
> While our most detailed geometric analysis focuses on BLIP-2, the appendix already includes cross-backbone results on **Qwen2-VL-7B-Instruct** and **Gemma3-4B**, where rank-1 Gap-Init consistently improves over random rank-1 and is competitive with or better than rank-8 baselines. On Qwen2-VL-7B-Instruct, rank-1 Gap-Init achieves 144.11 CIDEr vs. 143.67 for rank-1 random and 142.57 for rank-8 random. On Gemma3-4B, rank-1 Gap-Init achieves 95.24 vs. 90.35 / 89.62 under 1 epoch, and 96.17 vs. 95.43 / 94.82 under 5 epochs.
>
> So while broader validation on additional architectures would certainly strengthen the paper further, the current evidence already suggests that the effect is not confined to a single BLIP-2-style backbone. More broadly, we will revise the paper to clarify that our contribution is **not intended as a universal improvement for all LoRA settings**, but as a principled analysis and simple, geometry-aware initialization strategy for the **extreme low-rank multimodal PEFT regime**.
>
> Thank you again for the careful reading and constructive suggestions.

---

> > ### Author Rebuttal · Reviewer_EK4s · 2026-04-06
> >
> > The response addresses my concerns.

---

> > > ### Author Response · Authors · 2026-04-06
> > >
> > > Thank you for the update. We appreciate that the concerns are now fully resolved.
> > >
> > > Given this, we believe the remaining question is how to interpret the contribution. Our work is not a narrow rank-1 improvement, but a mechanism-oriented result that identifies a direction-limited failure mode in low-rank multimodal adaptation. Rank-1 serves as a diagnostic setting, while the underlying insight explains when and why low-rank methods depend on initialization. In this sense, the contribution extends beyond the specific setting studied.
> > >
> > > We would appreciate reconsideration of the overall rating in light of this. Thank you very much in advance!!

---

### Decision · Program_Chairs · 2026-04-30

**Decision:**

Accept (regular)

**Comment:**

This paper investigates why LoRA often fails or becomes unstable at the extreme limit of rank 1. The authors argue this is not due to the limited capacity of the LoRA adapter; instead, it is due to a modality gap that can be fixed with a proper initialization. The primary contribution is proposing a training-free initializer, called GAP-Init, that fixes this issue. They show experimentally that this quick fix makes rank-1 LoRA perform much better, and can even match higher ranks of up to 8. The method is simple to implement and shows big gains in performance and stability (lower variance in results). The authors included additional experiments in their rebuttal that would be useful to mention in the paper as well, such as extending the method to higher ranks.

However, while the paper is positive overall, it still has many limitations that are shared by many reviewers:

1. Performance seems to be sensitive to the calibration set. The authors show that pure out-of-distribution calibration degrades performance, but, on the positive side, they also show that mixing domains yields a slight improvement over the in-domain baseline.

2. the benefits of Gap-Init nearly vanish as you increase the LoRA rank, train for more epochs, or use larger, more modern MLLMs (like Qwen-7B), so it's unclear if it can be useful in practice.

3. This is specific to multimodal systems and it's unclear how to extend it to a single modality (e.g. text-only models).

4. authors mention in their rebuttal that the performance doesn’t seem to improve monotonically with the size of the calibration data that is used to estimate the modality gap, which is quite odd and suggests that one has to run a cross-validation to pick the best size of the calibration dataset.

For these reasons, my recommendation is weak accept.